# Citric Acid-Mediated Microwave-Hydrothermal Synthesis of Mesoporous F-Doped HAp Nanorods from Bio-Waste for Biocidal Implant Applications

**DOI:** 10.3390/nano12030315

**Published:** 2022-01-19

**Authors:** Gopalu Karunakaran, Eun-Bum Cho, Govindan Suresh Kumar, Evgeny Kolesnikov, Kattakgoundar Govindaraj Sudha, Kowsalya Mariyappan, Areum Han, Shin Sik Choi

**Affiliations:** 1Institute for Applied Chemistry, Department of Fine Chemistry, Seoul National University of Science and Technology (Seoul Tech), Gongneung-ro 232, Nowon-gu, Seoul 01811, Korea; 2Department of Physics, K.S. Rangasamy College of Arts and Science (Autonomous), Tiruchengode 637 215, Tamil Nadu, India; gsureshkumar1986@gmail.com; 3Department of Functional Nanosystems and High-Temperature Materials, National University of Science and Technology “MISiS”, Leninskiy Pr. 4, Moscow 119049, Russia; kea.misis@gmail.com; 4Department of Biotechnology, K.S. Rangasamy College of Arts and Science (Autonomous), Tiruchengode 637 215, Tamil Nadu, India; sudharhi@gmail.com (K.G.S.); kowsalyamariyappan23@gmail.com (K.M.); 5Department of Food and Nutrition, Myongji University, Myongji-ro 116, Cheoin-gu, Yongin 17058, Korea; gks0407@naver.com (A.H.); sschoi@mju.ac.kr (S.S.C.); 6Department of Energy Science and Technology, Myongji University, Myongji-ro 116, Cheoin-gu, Yongin 17058, Korea

**Keywords:** mesoporous nano-hydroxyapatite, citric acid, zebrafish toxicity, antimicrobial activity, implants applications

## Abstract

In this current research, mesoporous nano-hydroxyapatite (HAp) and F-doped hydroxyapatite (FHAp) were effectively obtained through a citric acid-enabled microwave hydrothermal approach. Citric acid was used as a chelating and modifying agent for tuning the structure and porosity of the HAp structure. This is the first report to use citric acid as a modifier for producing mesoporous nano HAp and F-doped FHAp. The obtained samples were characterized by different analyses. The XRD data revealed that F is incorporated well into the HAp crystal structure. The crystallinity of HAp samples was improved and the unit cell volume was lowered with fluorine incorporation. Transmission electron microscopy (TEM) images of the obtained samples revealed that a nano rod-like shape was obtained. The mesoporous structures of the produced HAp samples were confirmed by Brunauer–Emmett–Teller (BET) analysis. In vivo studies performed using zebrafish and *C. elegans* prove the non-toxic behavior of the synthesized F doped HAp samples. The obtained samples are also analyzed for antimicrobial activity using Gram-negative and Gram-positive bacteria, which are majorly involved in implant failure. The F doped samples revealed excellent bactericidal activity. Hence, this study confirms that the non-toxic and excellent antibacterial mesoporous F doped HAp can be a useful candidate for biocidal implant application.

## 1. Introduction

Bone regeneration is one of the most interesting fields in reformative medicine in which different calcium-dependent biomaterials are implemented for the recovery of bone [1]. The bone consists of bio-minerals along with its protein matrix, minerals, and water [2]. The bio-mineral component in bone is mostly hydroxyapatite (HAp) (Ca_10_(PO_4_)_6_(OH)_2_). The HAp has been applied to bone implants because of its similar chemical composition and increased bio-compatibility analogous to that of natural bone. HAp is one of the most stable phases of calcium phosphate and is found in several parts of our body, such as teeth, bone, etc. [3]. HAp is the most widely used material for different biomedical applications, such as dermal filler [4], wound healing [5], scaffolds [6,7,8], biomedical coating [9], tissue engineering [10], and bone regeneration application [11,12]. Even though HAp is widely used, its usage for surgical purposes in implantation is limited due to bacterial infection. To enhance the intrinsic antibacterial properties, several attempts have been made by incorporating antibiotics, functional nanoparticles insertion, antibacterial composite preparation, and metal ions incorporation.

In the HAp structure, different cations such as Ag^+^, Tb^3+^, Si^4+^, Fe^3+^, and Mg^2+^ can be replaced with Ca^2+^ ions, and anions like CO_3_^2−^, SO_4_^4−^, and fluorine can be replaced with hydroxyl bonds [13,14,15,16,17,18]. Among the anions, F^−^ reveals better physical and biological activities [19]. Fluoride is an important element that is essential for dental and skeletal formation [19]. Fluoride is found naturally in the bone and tooth enamel. To reduce the risk of dental caries, it is recommended that 1.5 to 4 mg/g of fluoride be consumed daily as a supplement [20]. It was revealed that the appearance of fluoride has essential activities in the development of bone [19]. It was found that F presence induces the proliferation of orthoclastic cells by altering the G-protein dependent-tyrosine phosphorylation pathway [21].

Fluoride is commonly known as a protective agent for the treatment of osteoporosis in the form of fluorapatite [22]. Stoichiometric fluorapatite has fluorine in the range of 3.77 wt.% but the fluorine range in human bone is lower than 1 wt.% [23,24]. Different researchers investigated different fluoridated apatite’s with varying degrees of fluoridation. F was discovered to increase crystal size due to the increased crystal development ratio, which also leads to a reduction in strain on the apatite and the integrity of the HAp shape. At lower degrees of fluoridation, the solubility of the HAp is reduced [25]. Some researchers have thoroughly investigated the mechanism by which fluoride reduces the solubility of hydroxyapatite [26,27]. Various synthesis approaches of F doped HA are known, like the mechanochemical process [21], sol-gel [28], the hydrothermal method [29], and precipitation [30]. Compared to the above-mentioned approach, the microwave method tends to be an effective method for fabrication of HAp in a fast way [31]. The microwave method is the top-class method for synthesis of the HAp [32]. The microwave method uses an electromagnetic spectrum ranging frequencies from 0.3 to 300 GHz and the wavelengths ranging from 1 mm to 1 m [33]. However, in the laboratory method, about 2.45 GHz is used as frequency. The photon energy frequency is generated at about 10^−5^ eV [34]. This photon energy is taken up by the reactant mixture (solute and solvents) at the molecular level, leading to the rapid heating up of the reactant mixture, which results in the formation of highly pure, uniform, and crystalline particles [34]. However, until today, the synthesis of mesoporous F doped HAp is not available using the microwave hydrothermal method. 

Mesoporous materials are gaining more importance nowadays due to their higher porous volume, unique surface area, higher biological behavior, and better biocompatibility [35]. Recently, it has been found that mesoporous HAp structures enhance bone recovery [35]. Hence, to improve the biological performance and wide range of applications, mesoporous HAp nanoparticle synthesis requirements have been increased. To produce a mesoporous structure, we aimed to use citric acid as a modifier. Citric acid is the most common weak acid which is naturally present in citrus fruits such as lemon, orange, kiwi, etc. [36]. It is widely used as a flavoring and chelating agent [36]. Citric acid is also used for the synthesis of nanoparticles. Recently, some researchers have used citric acid as a modifier, catalytic agent, and precipitating agent for the synthesis of nanoparticles such as copper nanoparticles, ceria nanoparticles, and tungsten trioxide nanoparticles [37,38,39]. Until today, there has been no research on the synthesis of mesoporous HAp nanoparticles using citric acid as a modifier. Hence, in this study, we aimed to utilize citric acid as a modifier for the production of F-doped HAp.

Furthermore, eggs are used in several food preparations, due to which, every year a million tons of bio-waste is generated in the world from eggs in the form of eggshells [40,41]. Hence, it is essential to reduce waste and find an alternative way to recycle this waste in a useful way. Thus, in this study, we aimed to utilize this bio-waste to produce useful F-doped HAp for biomedical applications. This study will work in dual ways. In one way, it will minimize the cost of F-doped HAp production. In another way, it will reduce environmental pollution. 

## 2. Materials and Methods

### 2.1. F-Incorporated HAp Obtained Via Bio-Waste Seashells

Citric acid was used as a modifier for obtaining HAp nanoparticles. Eggshells were used as a calcium precursor and potassium phosphate for the HAp synthesis. The synthesis of HAp was performed as: initially, the calcium precursor was prepared by taking 11.2 g of eggshell powder in the beaker and then 33.4 mL of HCl was added dropwise until the powder was dissolved and then it was filled to 100 mL using deionized water. The calcium precursor solution was further mixed with 100 mL of potassium phosphate (0.6 M) at 37 °C. After that, the pH of the solution was adjusted to pH 13 using potassium hydroxide. Then the reaction mixture was transferred to the reaction vessel and sealed well before microwave hydrothermal treatment. The MARS 6 Microwave Digestion System (CEM Corporation, Matthews, NC, USA) was used for the microwave irradiation at 700 W for a period of 30 min under controlled temperature of 200 °C and pressure at 300 psi. After the reaction is accomplished, the precipitate was collected and then further washed five times with water to remove the unreacted reagent. The final precipitates were rinsed twice using DD water and then dried at 90 °C for 24 h. After 24 h, the obtained white precipitate was crushed into a fine powder and then named pure HAp. For the formation of hydroxy fluorapatite, sodium fluoride (1, 3, 5 mol.%) was mixed to calcium suspension and the reaction was performed. The other procedure is similar to pure HAp synthesis. The dried powders were sintered at 90 °C for 12 h using an air atmosphere and applied for further identification.

### 2.2. Characterization

The eggshell was characterized by utilizing SEM-EDX (Tescan Vega 3—Brno, Kohoutovice, Czech Republic, SDD-X-Act Detector, Oxford Instruments Inc., Abingdon, UK) to analyze the morphology and elemental composition present in it. The crystalline phases of F-incorporated HAp samples (FHAp-1, FHAp-2, and FHAp-3) and pure HAp were determined by X-ray diffractometer (XRD) (Difray-401, Scientific Instruments Joint Stock Company, Saint-Petersburg, Russia) using chromium (Cr–*Kα*) (λ = 2.2909 Å) radiation as X-ray source. The crystalline phases and cell dimensions were identified to develop the diffraction angles. The cell parameters were determined by using the least square fitting method [42] via the UnitCell program tool. A Fourier (FT-IR) transform infrared spectroscopic examination was executed using the Nicolet 380 instrument (Thermo Scientific, Waltham, MA, USA) to estimate the molecular attachment and the alterations of cations and anions in HAp. The FTIR spectra were investigated between 400 and 4000 cm^−1^ of the procured powders in the pellet form added with spectroscopic grade KBr. Raman spectroscopy was carried out using the DXR Raman Microscope, Thermo Scientific, Waltham, MA, USA, to determine the functional groups and chemical composition. Further, the energy-dispersive X-ray (EDX) spectrum was recorded utilizing Tescan Vega 3—Brno–Kohoutovice, Czech Republic, SDD-X-Act Detector, Oxford Instruments Inc., Abingdon, UK to analyze the elements present in F doped HAp and pure HAp samples. In addition, the adsorption-desorption isotherms of nitrogen were performed for the prepared samples utilizing the Nova 1200e analyzer (Quantachrome Instruments, Boynton Beach, FL, USA) under relative pressure (*P/P_0_*) between 0.04 and 0.95. Based on the adsorption-desorption isotherms of nitrogen, pore size distribution [43], pore volume (*V*_t_), and surface area are determined. The structure and crystallinity of the selected area (SAED) electron pattern were characterized using the Transmission (TEM, JEM-1400, JEOL, Akishima, Japan) electron microscopy.

### 2.3. In Vivo Toxicity Estimation in Zebrafish Embryos and C. elegans

The *in vivo* toxicity of marked 25 numbers of zebrafish eggs dispersed in various ratios of (25–250 µg/mL) synthesized F doped HAp and pure HAp powders were assessed using the OECD-203 protocol and regulations. The embryos were added to 100 mL of Hank’s suspension along with different ratios of F-doped HAp and pure HAp powders. The embryos were incubated under controlled conditions and they were observed by using a 40× bright-field microscope at each 24 h period until 96 h for the generation of organs such as eyes, tail, heads, and liver [44]. For this assay, the water is regulated at a controlled temperature of 24 °C. The matured and dead embryos are measured separately according to the number of different solutions and all the samples are repeated in a triplicate manner.

The wild-type N2 strain of *C. elegans* was also used to investigate the safety of HAp particles by assessing worms’ movement and body size. Synchronized worms at larval stage (L4) were picked and placed on the nematode growth medium (NGM) agar plates containing bacterial food, *E. coli* OP50 with 200 μg/mL of pure HAp, and FHAp-3, respectively. After 12, 24, and 36 h of incubation, the number of body bends of *C. elegans* for 20 s was scored under the stereomicroscope (SZ61, Olympus, Tokyo, Japan). All measurements were carried out three times per group with more than 20 worms. To measure the body size of *C. elegans* after feeding them each type of HAp, the worms grown with particles until 1 day or 2 days of adulthood were moved to fresh NGM agar plates without bacterial lawn and particles. The body sizes of the individual *C. elegans* were measured using ImageJ software (http://imagej.nih.gov, accessed on 10 December 2019) after taking pictures of worms under the microscope connected to the camera (C-5050, Olympus, Tokyo, Japan).

### 2.4. Antibacterial Activity

The Gram-positive *Bacillus subtilis* (MTCC 1133) and Gram-negative *Pseudomonas aeruginosa* (MTCC 2581) bacteria are used for the determination of the antibacterial properties of obtained F doped HAp and pure HAp. The single inoculum of *P. aeruginosa* and *B. subtilis* from the Luria Bertani (LB) agar plate are added to the liquid LB culture suspension for the development of seed culture at room temperature for 24 h. After the seed culture was formed, it was swabbed over Agar plates (Muller-Hinton). The inoculated plates were loaded with disks of HAp as disk A (negative control), disk B (undoped pure HAp), disk C (FHAp-1), disk D (FHAp-2), and disk E (FHAp-3). After the disk loading, the plates were incubated for 24 h at a temperature of 37 °C. All the experiments were performed in triplicates and the zone of inhibition was analyzed after the completion of the incubation period. The antibacterial activity was determined based on the zone formation (inhibition zone) around the disk.

## 3. Results and Discussion

### 3.1. Elemental and Structural Characterization of Seashells, Pure HANRs, and Sr-Doped HANRs

The elemental and structural characteristics of the eggshell are shown in Figure 1a–c. The structural feature of the eggshell in Figure 1a–c shows that the eggshell consists of calcite crystals with an organic complex. Figure 1d represents the presence of elements such as O (59.3%), Ca (25.2%), C (15.1%), and Mg (0.4%) in the eggshell. Figure 2 shows the XRD pattern of prepared pure and F doped HAp samples. The 2*θ* diffraction peak values at 38.14° (002), 41.66° (102), 43.9° (210), 47.71° (211), 48.2° (112), 49.23° (300), 51.11° (202), 62.23° (310), 73.15° (222), 76.21° (312), 78.12° (213), 81.32° (104), and 83.35° (401), were detected for undoped pure HAp samples (Figure 2). The hexagonal crystallographic HAp phase is identified based on the above XRD results, which is also confirmed by the standard ICCD data file 09-0432 [45]. Furthermore, the 2*θ* diffraction peak values at 38.09° (002), 41.08° (102), 43.24° (210), 47.85° (211), 48.5° (112), 49.16° (300), 51.32° (202), 60.32° (310), 73.35° (222), 76.32° (312), 78.13° (213), 81.22° (104), and 83.42° (401) was detected for FHAp-1 samples (Figure 2). FHAp-2 and FHAp-3 also showed similar peak values to the FHAp-1 samples. The F^−^ ions are incorporated inside the HAp structure because of its unique replacing capacity with hydroxyl groups (OH^−^) from Ca_10_(PO_4_)_6_(OH)_2_ to Ca_10_(PO_4_)_6_ F_2_ [46,47]. The incorporation of F ions brings about changes in the HAp structure. The incorporation of F inside the HAp structure leads to a variation in the lattice parameters as well as the unit cell volume. The observed variation in the unit cell volume and lattice parameters are shown in Table 1.

The undoped pure HAp sample shows the lattice value as *a* = *b* = 9.2769 Å and *c* = 6.8712 Å. However, the FHAp-3 sample exhibits reduced lattice value as *a* = *b* = 8.6660 Å and *c* = 6.9864 Å. In addition, the pure HAp sample exhibits unit cell volume of 512.13 Å^3^, but the FHAp3 sample reveals reduction in the unit cell volume value of about 454.39 Å^3^. Addition of F leads to a reduction in the unit cell volume *V* value in FHAp3 up to 454.39 Å^3^. The FTIR peak of pure HAp, FHAp-1, FHAp-2, and FHAp-3 samples are revealed in Table 2 and Figure 3. The presence of PO_4_^3−^ group in pure HAp was observed at 471 cm^−1^, 559 cm^−1^, 601 cm^−1^, 964 cm^−1^, and 1010–1100 cm^−1^, respectively [48,49,50]. The occurrence of the broad O-H vibration (hydrogen bonding) due to the presence of water was observed at 3567 cm^−1^ [51,52]. Furthermore, the occurrence of water is observed at 1585 and 3330–3610 cm^−1^. The presence of the CO_3_^2−^ group was detected at around 860 and 1415 cm^−1^ [53,54]. The occurrence of the PO_4_^3−^ group was observed at 469–472 cm^−1^, 559–561 cm^−1^ and 600–603 cm^−1^, 961–964 cm^−1^, and 1010–1100 cm^−1^, respectively for the FHAp-1, FHAp-2, and FHAp-3 samples. The presence of the OH^−^ group was confirmed at 3570–3571 cm^−1^ and the CO_3_^2−^ group was confirmed at 861–869 cm^−1^ and 1402–1417 cm^−1^, respectively. The shifting of peaks from 1402 to 1417 cm^−1^ was detected; this may be due to the F inside the HAp structure.

The Raman spectra give a clear picture of the pure HAp formation and also of the doping of F in HAp, which is shown in Figure 4. The different modes from ν_1_–ν_4_ give detailed information about the exact changes in the undoped pure HAp and F doped HAp samples. However, the spectra (ν_1_) at 967 cm^−1^ confirms the occurrence of PO_4_^3−^ inside the pure HAp sample. Furthermore, spectra (ν_2_) at 415 cm^−1^ confirm the PO_4_^3−^ group double bending mode. The spectra at 1077 cm^−1^ and 607 cm^−1^ confirm the PO_4_^3−^ group triple (ν_3_) and tetra (ν_4_) stretching modes. Similar kinds of bending (ν_1_–ν_4_) modes were also observed for the HAp under different modes of synthesis [55]. Moreover, F-doped HAps (FHAp-1, FHAp-2, and FHAp-3) showed PO_4_^3−^ group different bending modes at 960, 1046, 591, and 428 cm^−1^, respectively. It was also observed that at 838 cm^−1^ an extra peak was observed for the FHAp-1 samples, which was observed to be a higher distribution for the FHAp-3 samples [56]. In addition, the FWHM of the peak due to ν_1_ was calculated and found to be 5.65, 5.55, 5.38, and 5.26 cm^−1^ for pure HAp, FHAp-1, FHAp-2, and FHAp-3 samples, respectively. A clear variation in the full width at half maximum of the ν_1_ peak is observed. This variation shows clearly that the F doping leads to the structural and crystallinity variations in the F doped HAp samples.

Figure 5 and Appendix A represents the elemental mapping and elemental spectra of the F doped and pure HAp samples. The observed result reveals that the distribution of carbon (C), phosphorus (P), oxygen (O), and calcium (Ca) were found in the HAp of pure samples. However, the F doped samples, namely FHAp-1, FHAp-2, and FHAp-3, show the good distribution of fluorine (F), carbon (C), phosphorus (P), oxygen (O), and calcium (Ca), respectively. The content of fluorine is different in all the F doped samples. Hence, the distribution of fluorine can be easily viewed in Figure 5. The lowest distribution of fluorine is seen in the FHAp-1 samples and the highest order of distribution of fluorine is observed in the FHAp-3 samples. The quantitative elements in fluorine-doped and the undoped pure HAp are revealed in Table 3. The major elements like phosphorus (P), oxygen (O), carbon (C), and calcium (Ca) were found in the undoped HAp samples. However, the occurrence of carbon (C), phosphorus (P), oxygen (O), fluorine (F), and calcium (Ca) was observed for the fluorine-doped HAp samples. In addition, the Ca/P contents ratio present in the samples like undoped pure HAp, FHAp-1, FHAp-2, and FHAp-3 represents 1.86, 1.67, 1.73, and 1.70, respectively. The Ca/P content ratio clearly indicates that the produced undoped pure HAp and fluorine-doped HAp samples are non-stoichiometric natured HAp. The F quantitative (Table 3) amount present in the F-doped HAp samples indicates 0.6, 1.2, and 3.2 at.% for FHAp-1, FHAp-2, and FHAp-3.

Figure 6a represents TEM analysis images of the undoped pure HAp samples which reveal the morphological features of the sample and it is found that the particles are nanorods with a size in the range of 7–10 nm width and about 20–30 nm length. The FHAp-1 sample in Figure 6c shows the particles with a mesoporous nanorod shape with a size range of about 5–8 nm width and about 10–30 nm length. The FHAp-2 sample in Figure 6e shows the particles with a mesoporous nanorod shape with a size range of about 4–8 nm width and about 8–25 nm length. The FHAp-3 sample in Figure 6g shows the particles with a mesoporous nanorod shape with a size range of about 3–7 nm width and about 6–20 nm length. The shape of the HAp is tuned with the (citric acid) organic modifiers and the microwave-mediated methods. In addition to the above observation, changes in the size of mesoporous nanorods also differ with the amalgamation of the F inside HAp structures.

Undoped HAp and F doped HAp SAED patterns are represented in Figure 6b,d,f,h. The acquired SAED patterns of the undoped HAp and F doped HAp are well matched with the XRD patterns. Hence, it confirms from both the results that the acquired powders are undoped HAp and F doped HAp nanorods without any impure phases in them. Usually, when phosphate combines with the calcium it forms amorphous structure due to the ionic interactions. However, when microwave treatment is given, rod-like structure is formed along the c-axis. Since citric acid is incorporated, it triggers the nuclei of HAp and leads to the formation of porous structure. Provas Pal demonstrated that citric acid is involved in the structural tuning of MnO_2_ and MnCO_3_ structures [57]. Citric acid changes the energy level between the molecules inside the HAp chain which facilitates porous structure formation.

The porous features and surface area of the undoped HAp and F-doped HAp were characterized by the nitrogen isotherm and the acquired results are shown in Figure 7a. The differences between the undoped HAp and F doped HAp samples can be easily identified by the nitrogen isotherm. Figure 7a reveals that all the samples consist of well-defined H3 type hysteresis loop IV isotherms, which is a clear confirmation of the porous characteristics of the materials [58,59]. Therefore, it is confirmed that the undoped HAp and F doped HAp samples are porous. However, the differences in the porous nature are figured out from the nitrogen isotherm graph. The Brunauer–Emmett–Teller (BET) surface area plot was obtained based on the obtained nitrogen isotherm graph and it is represented in Figure 7b. The undoped pure HAp, FHAp-1, FHAp-2, and FHAp-3 nanorods surface areas were about 58.3, 98.7, 113.0, and 117.8 m^2^ g^−1^, respectively.

The porous features such as pore diameter and pore volume of each sample are also identified and calculated based on the BJH-KJS (Barrett, Joyner, and Halenda—Kruk-Jaroniec-Sayari) method [43]. This method was introduced to measure the pore size distribution from the gas adsorption isotherm data using the assumption cylindrical pore geometry. The obtained graph by the above method is represented in Figure 7c. The undoped pure HAp, FHAp-1, FHAp-2, and FHAp-3 nanorods’ pore diameter was found to be 7, 7, 7, and 8 nm, respectively. However, the undoped pure HAp, FHAp-1, FHAp-2, and FHAp-3 nanorods’ pore volume was found to be 0.0377, 0.0761, 0.0551, and 0.1131 cm^3^g^−1^, respectively. It is clearly observed that with the increase in content of F in HAp, the pore diameter and pore volume are increased in HAp samples.

Porous structures play a key role in bone biomaterials and tissue engineering. It helps with surface attachment and fast recovery of the damaged bone and tissue [60]. Hence, to achieve the porous structure, we utilized citric acid as a chelating and structure tuning agent. Rui Chen et al. reported that citric acid leads to the formation of TiO_2_ porous structures [61]. The carboxyl and hydroxyl groups present in citric acid play a major role in porous structure formation. Gang Liu et al. also produced mesoporous aluminophosphate materials with the use of citric acid during the synthesis [62]. In another study, it was observed by Jianxin Zhou et al. that Ni-Mn oxide porous structures can be easily produced by the use of citric acid [63]. According to Yu Fan, citric acid plays a major role in maintaining the shape and structure of the H-form ZSM-5 zeolite [64]. Therefore, based on the above studies, it is clear that citric acid is involved in porous structure formation. Hence, in the present work, we utilized citric acid for the fabrication of porous HAp nanoparticles. The obtained porous structure will facilitate the fast recovery of the damaged bone or tissues.

### 3.2. In Vivo Toxicity Estimation in Zebrafish Embryos and C. elegans

The Zebrafish (*Danio rerio*) is one of the best models widely used to evaluate the toxicity of biomaterials due to its transparent features and it has similarities to the human genome [65]. The toxicity evaluation was performed based on the standard OECD guidelines. The undoped pure HAp, FHAp-1, FHAp-2, and FHAp-3 nanorods were added to Hank’s solution in various ratios, from 25 to 250 μg/mL. Earlier development of embryo formation is seen by using a microscope for the differentiation of dead and viable embryos. The *Daniorerio* reveals that the toxicological properties based on the mortality amount of sample exposed nanoparticles and the mortality amount was higher compared to 48 h.

Once the post-fertilization is completed the dead and live embryos were observed in the microscope. It was observed that the hatching rate of all the embryos is found to be enhanced with the pure and the F doped HAp samples. Figure 8 illuminates the mortality rate of 1.76% for undoped pure HAp and 1.96%, 2.1%, and 1.8% for the FHAp-1, FHAp-2 and FHAp-3 samples at 72 h. Even at the highest concentration, 250 µg/mL of FHAp does not show any toxic effect on the embryos. The observation of the tail, heart, eyes, and head was found to be normal without any effects by any HAp samples. Thus, it confirms that both undoped HAp and F-doped HAp samples are non-toxic towards zebrafish.

The soil nematode *Caenorhabditis elegans* has been considered as a model organism to investigate nanomaterials’ biological toxicity or safety due to its genetic and developmental similarities to humans. *C. elegans*’ body size and movement were used as indicators of manufactured nanomaterials’ toxicity [66,67]. When the wild-type N2 strain worms were cultivated with pure HAp and FHAp-3 powders from their egg stage, no significant changes in their body sizes and movements were found in early adulthood (≤2 days) in comparison with the control group that was cultivated without HAp (Figure 9 and Figure 10). Given that the worms’ eggs have experienced four developmental larval stages until the adult stage, these results demonstrate that those two types of HAp nanomaterials exert no developmental and biological toxicity on *C. elegans*.

### 3.3. Antibacterial Activity

The antibacterial properties of synthesized undoped pure HAp, FHAp-1, FHAp-2, and FHAp-3 nanorods towards Gram-negative *Pseudomonas aeruginosa* (MTCC 2581) and Gram-positive *Bacillus subtilis* (MTCC 1133) bacteria are effectively assessed and the obtained outcomes are revealed in Table 4 and Figure 11. The disk diffusion assay was implemented to conclude the antibacterial properties of obtained undoped pure HAp, FHAp-1, FHAp-2, and FHAp-3 nanorods towards bacterial samples. Images of bacteriostatic and zone of inhibition are revealed in Table 4 and Figure 11. The undoped pure HAp nanorods do not have bactericidal effects on *Pseudomonas aeruginosa* (MTCC 2581) and Gram-positive *Bacillus subtilis* (MTCC 1133) bacteria. The antibacterial properties of FHAp-1 nanorods against *Pseudomonas aeruginosa* and *Bacillus subtilis* were clearly observed. The obtained zone of inhibition (diameter of the spherical area) was 11.3 ± 0.57 mm towards *Pseudomonas aeruginosa* and 11.00 ± 0.10 mm towards *Bacillus subtilis.* The F content in HAp showed good antibacterial activity against the tested bacteria. Similarly, FHAp-2 and FHAp-3 nanorods also showed an increased zone of inhibition, such as 11.65 ± 0.11 mm and 12.30 ± 0.05 mm towards *Pseudomonas aeruginosa* and 12.03 ± 0.01 mm and 16.33 ± 0.11 mm towards *Bacillus subtilis*. Hence, it is clear from the observation that with the increase in F-doping, the antibacterial properties of F-doped HAp are effectively enhanced towards the tested bacteria. *Pseudomonas aeruginosa* and *Bacillus subtilis,* a general toxic pathogen that forms biofilms and is antibiotic-resistant, may cause bone implement infections in hospitalized patients [68].

The differences in the zone of inhibition were reflected with the Gram-negative *Pseudomonas aeruginosa* (MTCC 2581) and Gram-positive *Bacillus subtilis* (MTCC 1133) bacteria. The difference is due to the cell wall composition of different bacteria. The lysis of bacterial cells takes place with the involvement of nanorods in enzyme inhibition, cell wall lysis, oxidative stress, and inactivation of protein leads to cell death. The antibacterial potential of the nanorods is also subjected to their composition, morphology, and size. Supawadee Naorungroj et al. have observed that the fluoride content in the resin-based sealants has excellent antibacterial effects on *L. acidophilus* and *S. mutans* [69]. Another study by Monika Lukomska-Szymanska et al. has demonstrated that fluoride content in calcium fluoride plays an important role in antibacterial properties towards cariogenic bacteria such as *L. acidophilus* and *S. mutans* [70]. The highest antibacterial activity was found with 1.5 wt.% calcium fluoride. It was documented by Hamilton that fluoride plays an important role in acidogenicity in the dental plaque by inhibiting several enzymatic activities such as proton extruding ATPase, glycolytic enzyme inhibition, pyrophosphatase peroxidase inhibition, and catalase inhibition, which lead to bacterial cell death [71]. In another study, the growth inhibition effects of fluoride level on the plaque-causing mutants *streptococci* bacteria were tested on the glass ionomer [72]. The results confirmed that the mutans *streptococci* bacteria was found to be less with increased fluoride contents, which confirms its antibacterial effects.

A recent study by A. D. Anastasiou et al. clearly showed that fluorapatite has better antibacterial activity than strontium-doped fluorapatite against the tested bacteria such as *B. cereus, E. coli, B. subtilis*, and *S. aureus* [73]. Furthermore, it was also observed by Ahmed Alhilo et al. that hydrothermal mediated fluorapatite coating over stainless steel discs leads to excellent antibacterial activity against *A. actinomycetemcomitans*, *F. nucleatum*, and *P. gingivalis* [74]. The coating of fluorine enhances the surface roughness and lowers the surface charge, which leads to higher attachment and results in enhanced antibacterial properties by releasing fluoride ions. The latest report by Liyuan Zheng et al. showed that fluorine-doped zirconium oxide possesses excellent antibacterial effects through the dissociation of fluorine ions [75]. It was also found that fluorine-doped zirconium oxide also showed excellent antibacterial effects when incorporated as fillers in the composite resin for secondary caries treatments. The various research on fluorine applications clearly demonstrated that fluorine-doped materials possess excellent antibacterial properties and can be used for various biomedical applications. Hence, this fluoride-doped HAp can be used for implant coating applications.

Hence, the doping of F in HAp, the growth of *Pseudomonas aeruginosa,* and *Bacillus subtilis* are reduced effectively. Based on fluorine-mediated bone mineral formation, bone tumor suppression was excellent. The F doped HAp has significant importance in the fabrication of multi-properties of bone scaffold for scaffold materials. The maintained relationship between biocompatibility and antibacterial effects of F doped HAp in biomedical applications, i.e., the better cytocompatibility and yield properties, were achieved. Hence, this biomaterial can be effectively used as an implant coating material for enhanced antibacterial effects and better biocompatibility.

## 4. Conclusions

In conclusion, the fluoridated mesoporous nano-hydroxyapatite was obtained using citric acid as a modifying and chelating agent with the aid of microwave hydrothermal treatment. The observed characterization results confirm the formation of mesoporous nano-HAp and F-doped HAp. The changes in the crystal structure and shape of the HAp were also influenced by the content of F inside the crystal structure. The unit cell parameter and volume of the crystal structure were also found to be altered with the F incorporation. The in vivo analyses using zebrafish and *C. elegans* revealed that it was feasible to create better compatibility and less toxicity to the biological materials even at the higher concentration of F in HAp. In the end, the biocidal activity using bacteria concludes that the F doped HAp powder kills the bacteria well. The increased F ratio inside the HAp shows a higher degree of biocidal action towards bacteria that are mainly involved in implant-based infections. Thus, the non-toxic and higher degree of biocidal action of mesoporous F-doped HAp could be a promising, long-lasting biomaterial for biocidal implant application.

## Figures and Tables

**Figure 1 nanomaterials-12-00315-f001:**
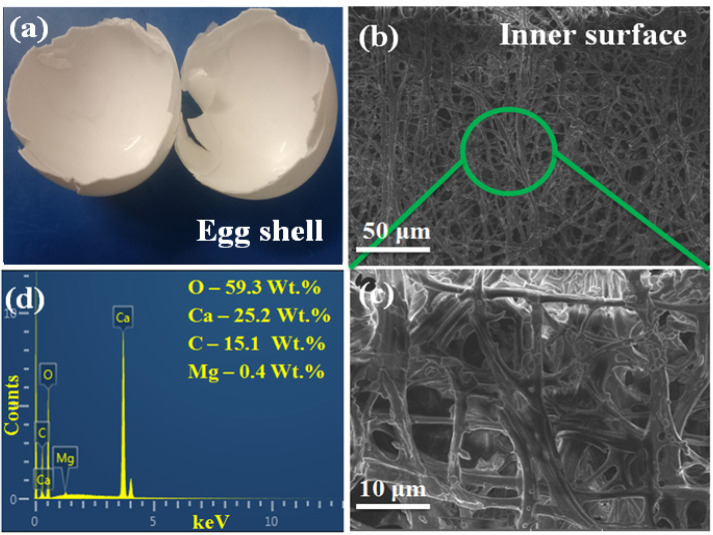
(**a**) Eggshell, (**b**,**c**) SEM image of cross section of eggshell under different magnifications, and (**d**) EDX spectrum of eggshell.

**Figure 2 nanomaterials-12-00315-f002:**
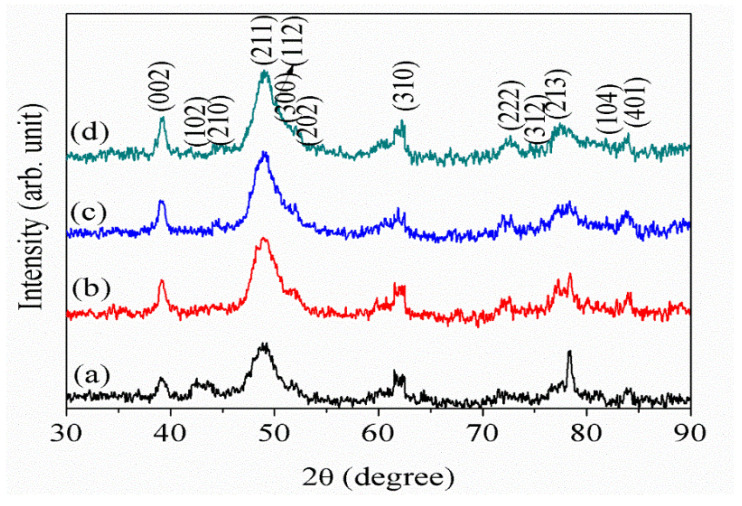
XRD patterns of (**a**) pure HAp, (**b**) FHAp-1, (**c**) FHAp-2, and (**d**) FHAp-3.

**Figure 3 nanomaterials-12-00315-f003:**
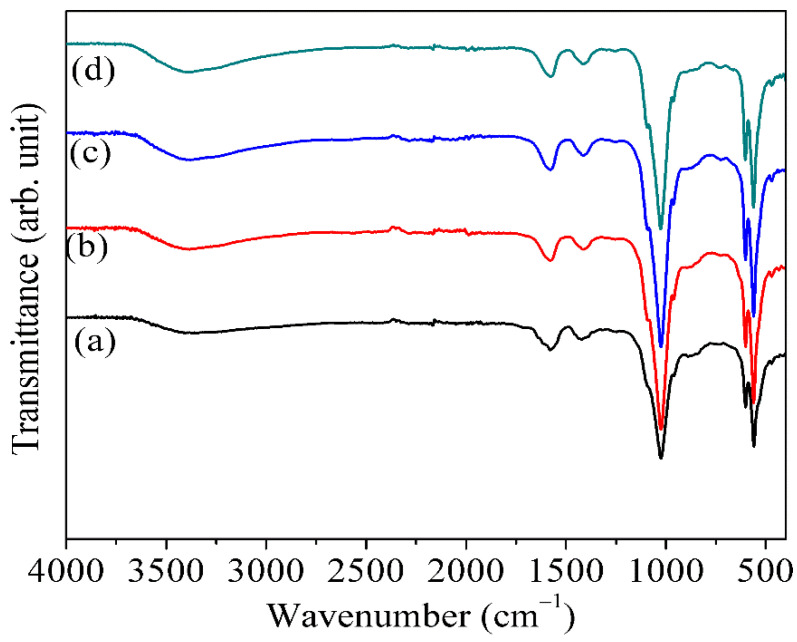
FTIR spectra of (**a**) pure HAp, (**b**) FHAp-1, (**c**) FHAp-2, and (**d**) FHAp-3.

**Figure 4 nanomaterials-12-00315-f004:**
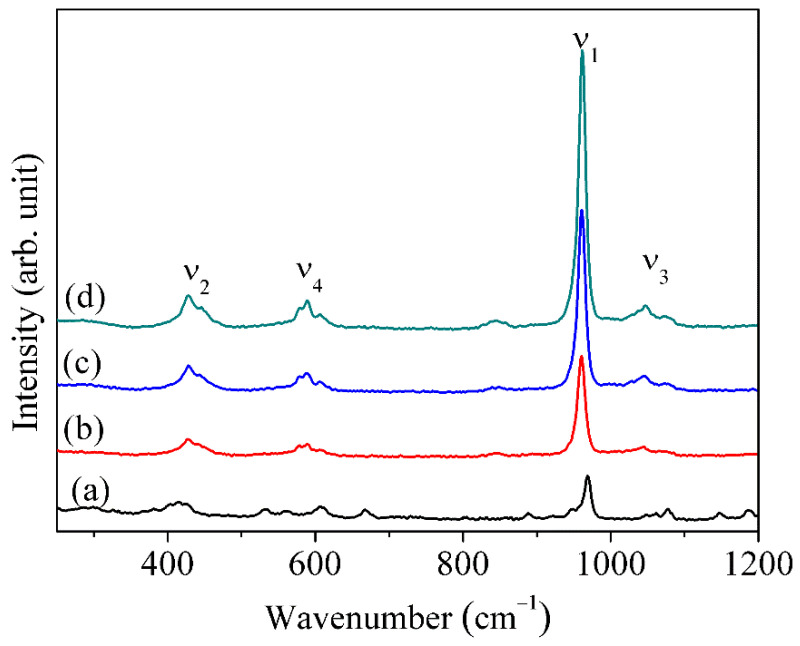
Raman spectra of (**a**) pure HAp, (**b**) FHAp-1, (**c**) FHAp-2, and (**d**) FHAp-3.

**Figure 5 nanomaterials-12-00315-f005:**
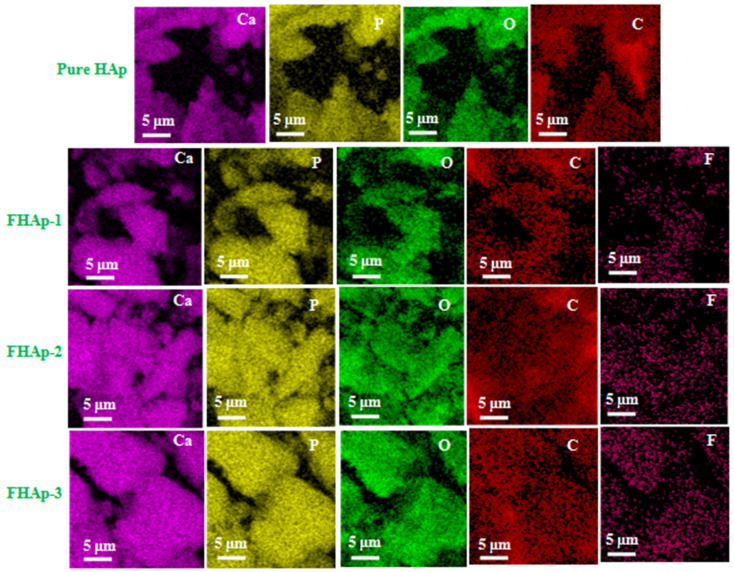
Elemental mapping of pure and F-doped HAp samples.

**Figure 6 nanomaterials-12-00315-f006:**
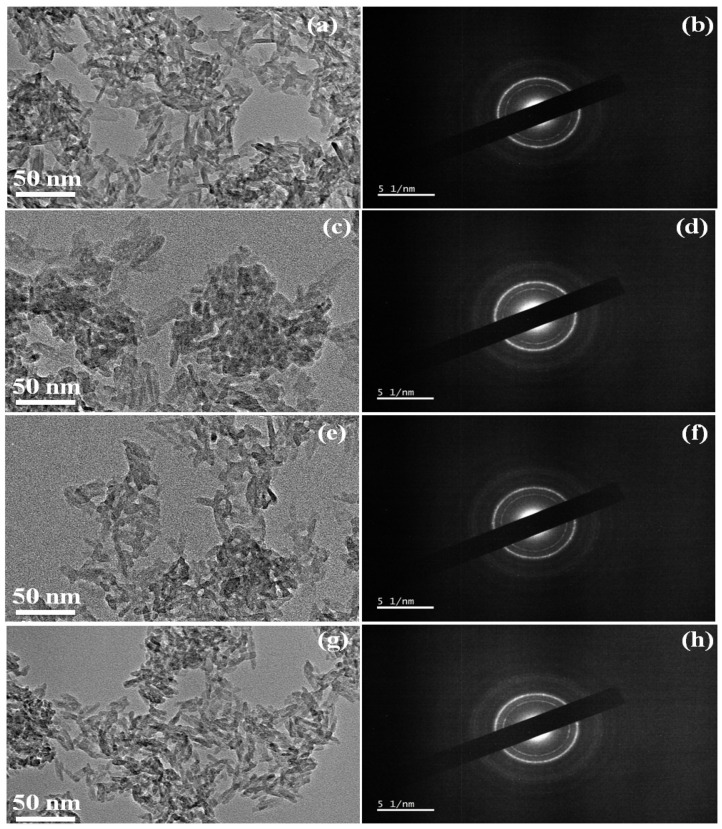
TEM image and corresponding SAED pattern (**a**,**b**) pure HAp, (**c**,**d**) FHAp-1, (**e**,**f**) FHAp-2, (**g**,**h**) FHAp-3.

**Figure 7 nanomaterials-12-00315-f007:**
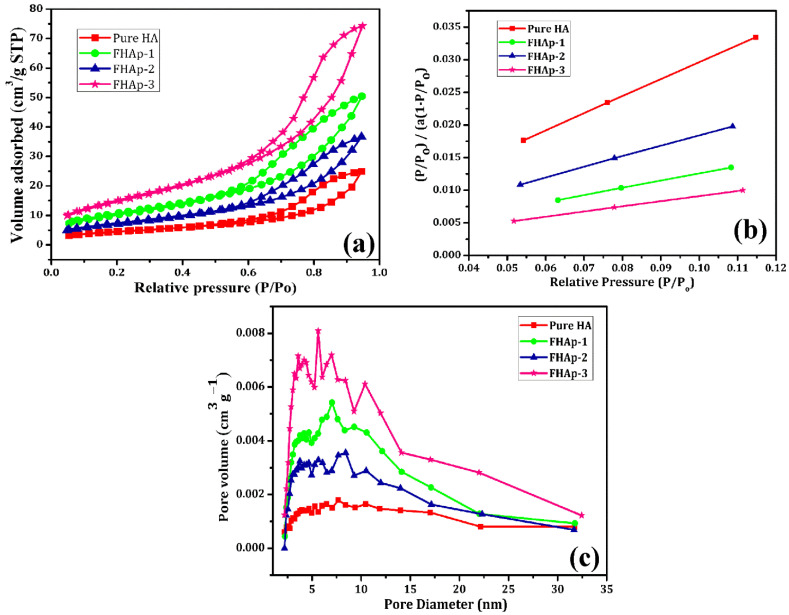
(**a**) Nitrogen adsorption-desorption isotherms, (**b**) BET line, and (**c**) BJH-KJS pore-size distributions of pure and F-doped HAp samples.

**Figure 8 nanomaterials-12-00315-f008:**
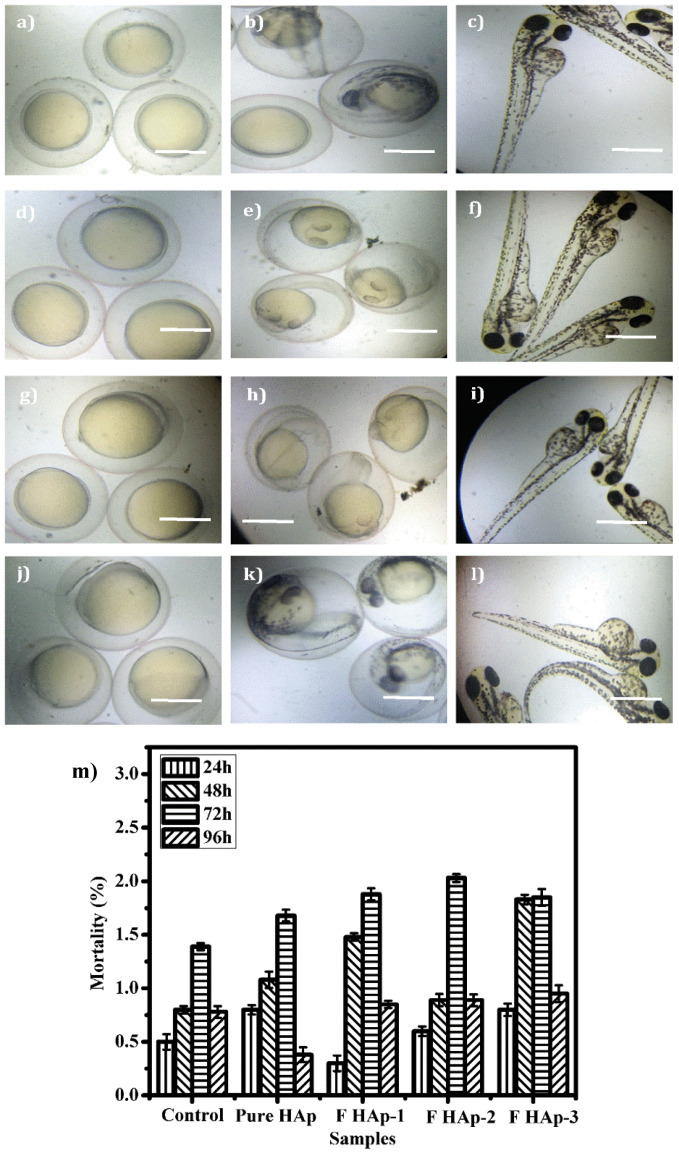
Images representing the Zebra fish embryos with hours post fertilization (hpf). Control after 24 hpf (**a**), 48 hpf (**b**), and 96 hpf (**c**). Pure HAp (250 µg/mL) treated after 24 hpf (**d**), 48 hpf (**e**), and 96 hpf (**f**). FHAp-1 (250 µg/mL) treated after 24 hpf (**g**), 48 hpf (**h**), and 96 hpf (**i**). FHAp-3 (250 µg/mL) treated after 24 hpf (**j**), 48 hpf (**k**), and 96 hpf (**l**). Bar graph (**m**) represents the mortality (death rate) % of prepared HAp with respect to time and concentration. Scale bars: 250 µm.

**Figure 9 nanomaterials-12-00315-f009:**
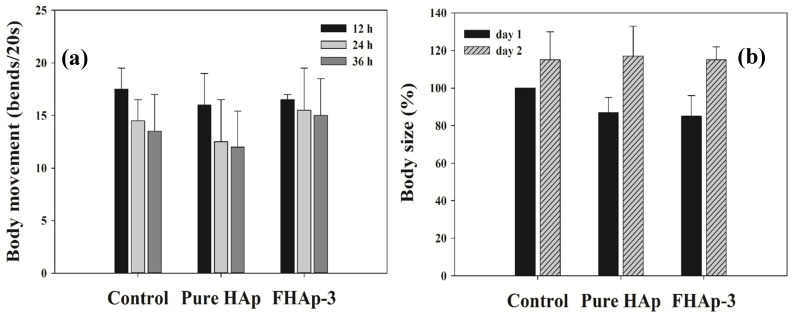
Safety assessment of pure HAp and FHAp-3 using *C. elegans* model. Both body movement (**a**) and body size (**b**) of worms fed with each type of HAp powder were not changed at the indicated times in comparison with the control group (without HAp). No significance was found between the control group and the worms fed with HAp groups through the statistical analysis (student *t*-test).

**Figure 10 nanomaterials-12-00315-f010:**
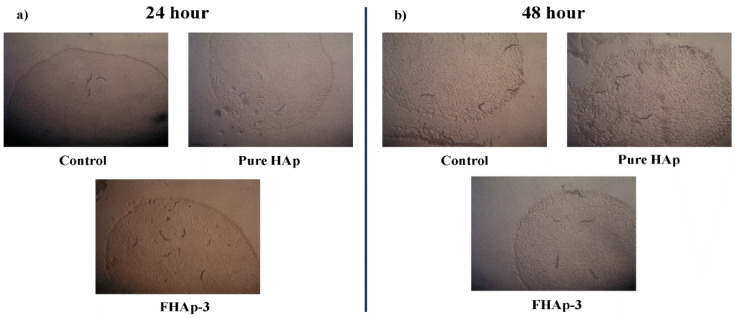
Effect of pure HAp and F-doped HAp on *C. elegans*’ behavior at different time intervals (**a**) 24 h and (**b**) 48 h.

**Figure 11 nanomaterials-12-00315-f011:**
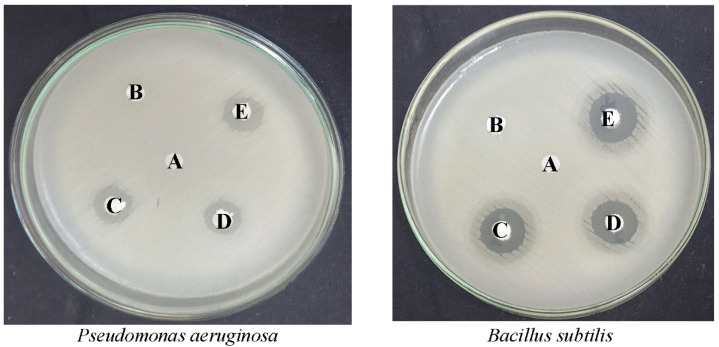
Antibacterial activity of (**A**) commercial HAp, (**B**) pure HAp, (**C**) FHAp-1, (**D**) FHAp-2, (**E**) FHAp-3.

**Table 1 nanomaterials-12-00315-t001:** Lattice parameters of crystalline HAp phase present in the synthesized samples.

Sample Code	Lattice Constants (Å)		Unit Cell Volume, V	Lattice Distortion (*c*/*a*)
	a = b	c	(Å^3^)	
Pure HAp	9.300 ± 0.20	6.871 ± 0.15	512.13 ± 0.14	0.7388
FHAp-1	9.302 ± 0.15	6.843 ± 0.20	512.84 ± 0.23	0.7356
FHAp-2	8.671 ± 0.30	6.970 ± 0.21	453.95 ± 0.35	0.8038
FHAp-3	8.666 ± 0.35	6.986 ± 0.22	454.39 ± 0.40	0.8061

**Table 2 nanomaterials-12-00315-t002:** FTIR spectral analysis of pure and F substituted HAp samples.

Sample Code	Wavenumber (cm^−1^)			
	Phosphate (PO_4_^3−^) Group	Hydroxyl (OH^−^) Group	Adsorbed CO_3_^2−^	Water (H_2_O)
Pure HAp	471, 559, 601, 964, 1010–1100	3567	860, 1415	1585, 3330–3610
FHAp-1	472, 559, 603, 961, 1010–1100	3570	869, 1417	1581, 3330–3610
FHAp-2	470, 560, 600, 963, 1010–1100	3571	862, 1409	1569, 3330–3610
FHAp-3	469, 561, 602, 964, 1010–1100	3570	861, 1402	1577, 3330–3610

**Table 3 nanomaterials-12-00315-t003:** Quantitative elemental composition of pure and F substituted HAp samples.

Sample Code	Elements (wt.%)				
	Ca	P	O	C	F
Pure HAp	32.6	17.5	44.3	5.6	--
FHAp-1	31.3	18.7	43.9	4.6	0.6
FHAp-2	31.5	18.2	43.8	5.3	1.2
FHAp-3	31.8	18.6	41.5	4.9	3.2

**Table 4 nanomaterials-12-00315-t004:** Antibacterial activity of HAp nanoparticles.

Microorganisms	Zone of Inhibition [Mean ± SD (mm)]			
	Pure HAp (100 µg/mL)	FHAp-1 (100 µg/mL)	FHAp-2 (100 µg/mL)	FHAp-3 (100 µg/mL)
*Pseudomonas aeruginosa* (MTCC 2581)	-	11.3 ± 0.57	11.65 ± 0.11	12.30 ± 0.05
*Bacillus subtilis * (MTCC 1133)	-	11.00 ± 0.10	12.03 ± 0.01	16.33 ± 0.11

## Data Availability

The data presented in this study are available on request from the corresponding author.

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
