# Peer review of "Citric Acid-Mediated Microwave-Hydrothermal Synthesis of Mesoporous F-Doped HAp Nanorods from Bio-Waste for Biocidal Implant Applications"

_nanomaterials, 2022, doi:10.3390/nano12030315_

Round 1
Reviewer 1 Report
G. Karakunaran present the interesting synthesis of fluorine-doped hydroxyapatite nanorods and study some of their properties like toxicity and antibacterial effects. I have two principal questions:
1.) My main question is why the authors do not mention the details of the microwave synthesis. Exactly the details on which the whole paper stands are missing (reaction time, temperature, type of microwave reaction vessel)
2.) Although the paper presents the good toxicity in models and the bactericidal effect, the authors do not present a direct application of the nanorods, where they also demonstrate this capacity. How do you envision the integration of the nanorods into artificial bone material? Or into coatings? How do the nanorods mix with the commonly used materials?
Therefore I recommend major revision.
Please find also some minor comments in the attached pdf.

Reviewer 2 Report
This paper shows a study of the effects of F doping in mesoporous nano- HAp fabricated through a citric acid-enabled microwave hydrothermal approach. Although a series of techniques were applied to investigate the change in the structure induced by F doping, including XRD, FTIR, Raman and EDX analyses, the incorporation of F in HAp to replace OH lacked a clear experimental support, and the formation of pure HAp without the presence of other calcium phosphate, i.e., nonstoichiometric HAp, needed further confirmations.
- For example, the authors claimed a structural distortion was caused by F doping from XRD measurement, but it was hard to draw such a conclusion due to the poor quality of the XRD curves and a missing of detailed information. The authors should explain why the XRD peaks in Fig. 2 and their assignments are completely different from those reported in literature (The excitation source should be given). Which peaks were used for calculating the lattice parameters? Error analysis of the values given in Table 1 should be performed. To be honest I could find the difference among the samples, taking into account the noise shown in the curves.
- Also, the authors claimed “The 234 shifting of peaks from 1402 to 1417 cm-1 in FTIR spectra was detected due to the insertion of F inside the 235 HAp structure.” However, the signal is too weak to give this conclusion.
- Could the authors explain why the v1 peak position in the Raman spectra changed from 967 cm-1 (give in text, but 964 cm-1 in table 2) for pure HAp to 961 cm-1 for FHAp-1 but to 964 cm-1 again for FHAp-3, if it were caused by F-doping? The observations seemed to demonstrate a nonstoichiometric nature of the obtained HAp samples.
- The authors gave the EDX elemental maps in Figure 5, but the EDX spectra were missing. The spectrum can give a direct support if the F signal is observable.
One suggestion: XPS can also be a good tool (F1s) to quantitatively examine the presence of F.
The most significant changes induced by F doping (?) were the variations of pore size and distribution given in Figure 7, and the antibacterial activity of the F doped samples shown in Fig. 11, which made this paper deserve publication. However, if these changes were associated with F incorporation but not other factors, an explanation/discussion is needed.
Based on above, this paper needs a substantial revision.
Round 2
Reviewer 1 Report
Thank you for responding to my questions and adding more experimental details!
I recommend acceptance and added just a few minor comments in the attached pdf.

Reviewer 2 Report
The authors have improved the quality of the manuscript. However, the irregular shift of the v1 peak position in the Raman spectra indicated that it is hard to claim that the structural change was caused only by F-doping. Please check the full width at half maximum of the v1 peak (influenced by structural alteration and crystallinity) to find whether there is a similar trend of variation with F content.
Accordingly, taking into account the key role of porous structure, a question arises: how could the authors claim there is a relationship between the porous structure and the incorporation of F in HAp, if there is no any explanation or discussion? Moreover, did this porous structure (or surface roughness) significantly influence the antibacterial activity shown in Fig. 11 (e.g., the variations of zone of inhibition with F content shown in Table 4)?
As mentioned in my previous comment, the most important changes were the variations of pore size and distribution given in Figure 7, and the antibacterial activity of the F doped samples shown in Fig. 11, but an explanation/discussion is needed to show these changes were associated with the content of incorporated F but not other factors.
Based on above, this paper still needs a revision before it can be accepted.
Round 3
Reviewer 2 Report
It has been improved
.